# The Effect of Chemical and Thermal Treatment for Desizing on the Properties and Chemical Functional Groups of Carbon Fiber

**DOI:** 10.3390/ma16206732

**Published:** 2023-10-17

**Authors:** Kyungeun Kim, Minsu Kim, Gyungha Kim, Daeup Kim

**Affiliations:** 1Korea Institute of Industrial Technology, Jeonju-si 54853, Republic of Korea; kke@kitech.re.kr (K.K.); mskim85@kitech.re.kr (M.K.); 2Department of Carbon Material Fiber Engineering, Jeonbuk National University, Jeonju-si 54896, Republic of Korea

**Keywords:** recycled carbon fiber, desizing, acetone treatment, heat treatment, functional group, mechanism

## Abstract

In this paper, in order to upcycle carbon fibers (CF), the changes in their mechanical and chemical properties in accordance with time and temperature were investigated, in addition to the oxygen functional group mechanism. When acetone as a chemical desizing agent was used, treatment with acetone for 0.5 h at 60 °C was the optimal condition for the complete removal of the sizing agent, and there was no deterioration in tensile strength. At 25 °C, the carbonyl group (C=O) and hydroxyl group (C-O) declined in comparison to commercial CF, but a novel lactone group (O=C-O) was created. At 60 °C, the oxygen present in the sizing agent was removed and C=O, C-O, and O=C-O decreased. On the contrary, in the case of thermal desizing in an inert gas nitrogen atmosphere, by increasing the temperature, functional groups combining carbon and oxygen were reduced, because nitrogen and oxygen atoms combined with C=O and C-O on the CF surface were eliminated in the form of CO, NO, CO_2_, NO_2_, and O_2_. When desizing via chemical and thermal methods, the amount of functional groups combining carbon and oxygen on the CF surface decreased. Desizing was performed as a pretreatment for surface treatment, so the methods and conditions were different, and related research is insufficient. In this study, we attempted to derive the optimal conditions for desizing treatment by identifying the surface characteristics and mechanisms according to chemical and thermal desizing treatment methods.

## 1. Introduction

Carbon fiber, which has advantages such as a low density, high specific strength, and chemical stability, is applied to high-tech industries and mostly used in aerospace, national defense, and sports cars [1,2,3,4,5,6,7,8,9]. However, the high price of CF and its expensive manufacturing process make it unreasonable to expand and apply CF to all industrial fields (e.g., commercial vehicles). In addition, thermosetting resin-based carbon composites, which are mainly used for aircraft and vehicle parts, are difficult to recycle and mostly treated as waste in the form of landfill and incineration, causing environmental pollution. To expand the application of carbon composites, upcycling technology that recovers and recycles end-of-life carbon composites is absolutely necessary to lower CF prices and mitigate environmental problems [8,9,10,11]. Currently, methods for retrieving recycled carbon fibers (rCF) from used carbon composites are being developed [12,13,14,15], but there are limitations to reusing recovered rCF, directly owing to the presence of impurities from the sizing agent and the separation process. In general, commercial CFs are sized with epoxy, which is a thermosetting resin, to improve their interfacial bonding strength with the resin, and when a thermoplastic resin is used, a desizing process is required to achieve a sizing treatment that harmonizes with the resin [16,17,18]. In addition, when wet-laid nonwoven fabrics are used to produce carbon composite parts via compression molding, the dispersibility of CF in which the sizing agent is not removed in water deteriorates due to tangling and agglomeration between the fibers, adversely affecting the properties of the nonwoven fabric [18,19,20]. Therefore, the desizing of CF is an essential process to improve the wettability when mixing CF with thermoplastic resins and manufacturing wet-laid nonwoven fabrics. Desizing treatment is a method that can improve wettability with fluids by exposing the surface of the CF and forming functional groups containing oxygen while removing the sizing agent. Moreover, it is carried out before the surface treatment, which generally involves chemical treatment methods [21,22,23,24,25,26,27,28,29] using organic solvents (e.g., acetone and ethanol) and thermal treatment [29,30,31,32] methods. In the case of using acetone as a chemical treatment method, the treatment temperature varies from room temperature to 70 °C, and the treatment time varies from 10 min to 24 h, depending on the researcher, so there is no consistency in the optimal treatment conditions [22,23,24,25,26]. Kim et al. reported that, after contaminants were removed through the ultrasonication of the sample in ethanol for 10 min, intense pulsed light (IPL) treatment improved the surface modification and chemical adsorption (O/C) of carbon-fiber-reinforced plastic [27]. Jang et al. also reported that CFs desized by immersion in dichloromethane for 5 days and distilled water for 2 days were treated with plasma and nitric acid in an oxygen atmosphere to improve the interlaminar shear strength (ILSS) and flexural strength of composites combined with polybenzoxazine resin, due to the increased surface roughness of the CFs [28]. In addition, Ibarra et al. reported that micropores increased on a specific surface of CF in line with the nitric acid treatment time after treatment in a tetrahydrofuran (THF) solution for 24 h using the Soxhlet method. It was confirmed that the mechanical strength decreased [29]. On the other hand, in the case of thermal desizing, Ahmed et al., described that, when heat treatment was carried out in an oven at 380 °C for 1 h, the epoxy on the CF surface was removed and oxidized to CO_2_ and water vapor [30], and Liu et al. showed that CF/PEEK composites fabricated by resizing polyetherimide (PEI) to CF subjected to desizing in an oven for 30 min increased the ILSS value by 16.1% compared to composites without sizing treatment [31]. According to Jiqiang et al., the surface roughness of CF washed with acetone after heat treatment at 500 °C for 5 h in a nitrogen atmosphere increased by about 36% compared to that of untreated CF, which improved the wettability with resin because of the increase in the surface area of the fiber, which confirmed that the interfacial bond strength increased [32]. In the case of producing wet nonwoven fabrics with CF, Choi et al. used CF without sizing treatment and confirmed the dispersion effect according to the type of dispersant and binder. As a result, it was confirmed that the agglomeration of the CF was reduced and its dispersibility in water was improved when a polyacrylic acid-based dispersant and polyvinyl alcohol-based binder were used [33]. Thus far, research has indicated variations in the pretreatment method for CF surface treatment, and the mechanism behind the changes in surface properties and chemical structure during the desizing treatment of CF remains unclear. In this study, for the purpose of establishing an upcycling technology for rCF recovered from carbon composites, chemical and thermal desizing treatments were performed to derive the optimal desizing conditions, and their effect on the changes in the mechanical and chemical properties of CF was reviewed. Furthermore, the chemical state changes and mechanisms of the CF surface relative to the desizing process conditions were found out.

## 2. Experimental Details

The Toray rCF used in this paper was recovered from an automobile fuel tank. The characteristics of the rCF and commercial CF were compared and are shown in Table 1. The tensile properties of the rCF were about 20% worse than those of the commercial CF, but it was confirmed that the surface properties of the CF were thermally and chemically stable [34]. During the desizing treatment, acetone (99.5%, Daejung Chemical, Siheung, Korea) was used as a solvent for chemical treatment. A process of immersion in a beaker containing acetone and a method of circulating cold water by connecting a reflux condenser device to a round flask containing acetone and carbon fiber were used. This is a device in which the solvent vapor was cooled and liquefied by circulating cooling water through the central glass tube and returned to the container. It was used as a method of extracting the carbon fibers by heating acetone. It was desized according to the temperature (25–60 °C) and treatment time (0.5–2 h) using an immersion and reflux condenser, and then dried at 100 °C for 1 h. The desizing treatment via heat treatment was conducted under the subsequent conditions: a treatment time of 0.5 h, a temperature increase rate of 5 °C/min, and a flow rate of 200 sccm at 300–1000 °C in a nitrogen (99.99%) atmosphere.

The surface changes in the commercial CF and desized treated CF were scrutinized under an accelerating voltage of 20 kV using Field scanning emission electron microscopy (FE-SEM). For the evaluation of the mechanical properties of the CF, as specified by the ASTM D3822 standard, a short fiber tensile test was performed, and the average value was obtained by performing the test more than 25 times per test condition at a speed of 5 mm/min. The amount of sizing removed was analyzed through a thermogravimetric analysis (TGA), and the temperature was increased to 1000 °C under the conditions of a 10 °C/min heating rate in a nitrogen atmosphere. The surface elements and chemical functional groups of the CF were detected using X-ray photoelectron spectroscopy (XPS, Nexsa). The test piece was irradiated with monochromatic Al Kα (1486.6 eV), and the high-resolution spectrum was acquired under the condition of a beam size (400 μm) and pass energy (50 eV). In addition, using the Wilhelmy plate method, diiodomethane and water were dropped on the CF at an injection rate of 6 mm/min, and the contact angle was measured to calculate the surface energy. This angle was measured above three times for the respective conditions, and the average value was calculated.

## 3. Results and Discussion

### 3.1. Characteristic Change of Carbon Fiber According to Desizing Process Conditions

The surface of the CF following the desizing process was observed and is shown in Figure 1. In the case of desizing with acetone, a change in the CF surface in terms of conformity with time and temperature could not be observed. On the other hand, when using thermal desizing in a nitrogen atmosphere, no surface defects appeared on the CF up to 500 °C, but it was confirmed that the surface of the fiber was harmed at 1000 °C. Ibarra et al. confirmed that the desizing treatment using THF was smooth without significant changes on the CF surface and that there were almost no defects [29], and Kim et al. found that CF reacts with oxygen in the atmosphere at above 500 °C when it is heat treated in an oxygen atmosphere. It was also reported that surface defects were generated and then partly disappeared, along with a decrease in the diameter of CF from above 600 °C [34]. In the present study, it was considered that desizing with acetone does not damage the surface of CF because of low energy. On the contrary, when thermal desizing in an inert gas nitrogen atmosphere, the surface damage of CF was confirmed at 1000 °C, which is higher than 500 °C [34], where defects on the CF surface were observed in an oxygen atmosphere. During heat treatment, the atmosphere can be seen to have a greater effect on the surface reaction of CF than the heat treatment temperature. It was determined that the energy that damages the bond is small [34]. 

In contrast, Figure 2 shows the tensile properties evaluated on the basis of the chemical and thermal desizing treatments. The tensile strength, modulus of elasticity, and elongation of the CF showed a similar trend. In the case of desizing with acetone, there was little difference in the tensile properties from those of the commercial CF as the time and temperature increased. In contrast, when thermal desizing, the tensile properties up to 500 °C were almost the same as those of the commercial CF within the error range, but at 1000 °C, the tensile strength rapidly decreased by about 70% to 1.26 GPa compared to the commercial CF. According to a study by Lee et al., it was reported that, after 1 min of plasma treatment, the surface of the fiber is eroded, making tensile strength measurements impossible [21]. It was confirmed that the tensile strength value gradually decreased from above 400 °C, and, at 600 °C, the CF deteriorated to an extent that an evaluation of its tensile properties was impossible and the mechanical properties decreased [34]. As a result of desizing with acetone within the scope of this study, it is considered that there were no defects on the CF surface and no change in the tensile properties due to the low energy application. However, in case of thermal desizing in a nitrogen atmosphere, the temperature at which the tensile properties of the CF deteriorated was increased compared to that in an oxygen atmosphere. This result is believed to have been due to the slowed down rate of CF degradation due to oxidation.

The change in the amount of residue on the CF surface, depending on the desizing process, was confirmed using a TGA. Figure 3 shows the graphs of the TGA and DTG results according to the desizing process. A weight loss of 0.12% was confirmed in the rCF before the desizing treatment, which is believed to have remained on the CF surface when separating the CF from the used carbon composites, requiring an additional desizing treatment. On the other hand, as a result of the desizing with acetone to optimize the desizing conditions in the commercial CF with an epoxy sizing of about 1%, the weight decreased by 0.21% at 25 °C, but no weight change was observed at 60 °C. Through this finding, it was confirmed that some of the sizing agent remained in the CF treated with acetone at 25 °C. Furthermore, at 60 °C, near the boiling point of acetone, the sizing agent was completely removed at about 320 °C, which is a decomposition temperature of the epoxy-based sizing agent coated on the surface. When desizing with acetone, 0.5 h at 60 °C was judged to be the optimal condition. On the contrary, in the matter of thermal desizing in a nitrogen atmosphere, desizing was performed under all conditions, and the results are representatively shown at 300 °C in Figure 3c,d.

### 3.2. Changes in Chemical Properties of Carbon Fibers Depending on Desizing Process Parameters

Figure 4 shows the C1s and O1s XPS spectra for analyzing the chemical changes in the CF treated according to the desizing process. A look at the C1s spectra of the CF desized with acetone reveals that the change from 25 °C to 60 °C was similar within the error range, and compared to the commercial CF, the carbonyl group (C=O) and hydroxyl group (C-O) were significantly reduced, while the lactone group (O=C-O) was elevated. In the O1s spectra, compared to the commercial CF, C-O decreased, but O=C-O increased. In contrast, the C1s spectra results of the thermally desized CF in an inert gas nitrogen atmosphere tended to be similar to those of the chemical treatment, with C=O and C-O decreasing significantly in comparison to commercial CF, especially at 1000 °C. O=C-O was created until 300 °C; however, it then decreased after 500 °C. In the case of the O1s spectra, in the results of thermal desizing up to 1000 °C, the quantity of C-O gradually declined in comparison to the commercial CF, and O=C-O increased until 500 °C, but declined at 1000 °C. Table 2 summarizes the O/C that can quantitatively determine the degree of composition change and oxygen content increase on the CF surface. The composition changed after the treatment with acetone up to 60 °C, carbon and nitrogen increased while oxygen decreased compared to the commercial CF, and silicon increased slightly at 25 °C and decreased at 60 °C. Conversely, in the case of thermal desizing in a nitrogen atmosphere, it was indicated that the amount of carbon tended to increase compared to the commercial CF by raising the temperature to 1000 °C, the amount of oxygen decreased until 1000 °C, and the amount of nitrogen increased up to 500 °C. It declined at 1000 °C. The CF interfacial bonding strength can be confirmed by the ratio of O/C [35], and, when desizing with acetone, the ratio decreased as the temperature rose to 60 °C compared to the commercial CF. On the other hand, the thermal desized CF decreased according to the temperature increasing up to 1000 °C, similar to the chemical desizing treatment. In particular, it decreased significantly to 0.11 at 1000 °C, showing the lowest value compared to the commercial CF. This is lower than the surface activity standard of 0.14 [35], and it was judged that the surface activity was poor at 1000 °C, resulting in a decrease in interfacial bonding. 

To check, in detail, the change in functional groups combining carbon and oxygen on the surface of the CF, the separated C1s peak is exhibited in Figure 5 and Table 3. When desizing with acetone, compared to the commercial CF, C-C, C=C, and C-N were elevated up to 60 °C, but C=O and C-O decreased, and O=C-O increased at 25 °C and then decreased at 60 °C. On the other hand, when using thermal desizing, C=C and C-C increased up to 1000 °C compared to the commercial CF, but C=O and C-O decreased, and C-N decreased up to 500 °C and then increased at 1000 °C. O=C-O increased up to 500 °C but decreased at 1000 °C. Lee et al. reported that oxygen increased after 1 min of plasma treatment, resulting in an increase in functional groups combining carbon and oxygen [21]. Additionally, according to Kim et al., after heat treatment in an oxygen atmosphere, the O=C-O bond inside the CF was greatly improved at 500 °C, and the functional groups combining carbon and oxygen increased, which was expected to improve the interfacial bonding force with the resin [34]. It was reported that, when the surface was treated in nitric acid, this resulted in an increase in functional groups of carbon and oxygen [36]. Because of this, desizing CF using acetone as the sizing agent on the CF surface that is removed by acetone at 25 °C, C=O and C-O are reduced, and it is judged that the remaining sizing agent in a small amount is converted into O=C-O by bonding oxygen. After that, at 60 °C, the sizing agent is completely removed, the bond with oxygen bonded to the CF surface end is broken, and it is considered that the C=O, C-O, and O=C-O bonds are reduced. On the other hand, as the temperature increases up to 1000 °C, the oxygen of C=O and C-O existing in the CF combine with nitrogen atoms and are removed as NO, NO_2_, and O_2_ by thermal energy, and it is judged that C-O and C=O decrease [34].

To confirm the change in the surface free energy of the CF according to the desizing process, the contact angle measured using hydrophobic and hydrophilic solutions is shown in Figure 6. The nonpolar and polar surface free energy values were used by substituting the following Equation (1).
(1)$$\frac{\gamma_{L}(1+\cos\theta )}{2{\left(\gamma_{L}^{D}\right)}^{\frac{1}{2}}}={\left(\gamma_{S}^{P}\right)}^{\frac{1}{2}}\times {\left(\frac{\gamma_{L}^{P}}{\gamma_{L}^{D}}\right)}^{\frac{1}{2}}+{\left(\gamma_{S}^{D}\right)}^{\frac{1}{2}}$$

The contact angle of the CF desized with acetone was similar to that of the commercial CF at 25 °C, and slightly increased at 60 °C. In contrast, thermal desizing in a nitrogen atmosphere was similar to that of the commercial CF at 300 °C, but gradually increased as the temperature increased to 1000 °C, and then increased by about 40% at 1000 °C. Figure 6 shows the contact angle results, and the surface energy was divided into polar and non-polar. As a result of the desizing with acetone (Figure 7a), the surface energy tended to decrease slightly as the temperature increased up to 60 °C, and the polar/surface energy ratio showed little change according to the treatment temperature, being about 30% lower than the commercial CF at 60 °C (Figure 7c). On the contrary, as shown in Figure 7b, in the result of the thermally desized CF, there was no significant difference to the commercial CF up to 500 °C, but it tended to decrease at 1000 °C. Figure 7d shows the polar energy/surface energy ratio also rapidly decreased after 500 °C and showed the lowest value of 0.23% at 1000 °C. From these results, it is determined that, when desizing with acetone at 60 °C, the functional groups combining carbon and oxygen shown in the sizing agent on the surface of the CF are removed, thereby reducing the amount of oxygen on the surface of the CF, reducing the polar surface free energy but increasing the contact angle. On the other hand, in the case of thermal desizing, as the temperature increases up to 1000 °C, the degree of damage to the surface of the CF gradually increases, so the C=O and C-O present in the CF decrease, and the surface of the CF decreases. It is believed that the bond between the carbon and oxygen present is broken, reducing the polar energy and increasing the contact angle.

### 3.3. Changes in Chemical Properties of Carbon Fibers Depending on Desizing Process Parameters

Figure 8 illustrates a schematic of the chemical structure and functional groups combining carbon and oxygen mechanism of the CF according to the results of analyzing the chemical and mechanical properties of CF based on the chemical and thermal desizing process. When desizing with acetone as a chemical treatment method, the sizing agent and impurities existing on the CF surface were removed at 25 °C, C=O and C-O declined, and some remaining sizing agent combined with oxygen increased the O=C-O. At 40 °C, the bonds of a small amount of residue on the surface of the CF were broken and converted into C=O and C-O bonds, and the surface of the CF treated at 60 °C was considered to be converted into C=O and C-O by combining the O=C-O bonds broken by thermal energy and the carbon atoms on the surface of the CF as the sizing agent was completely removed. On the other hand, thermal desizing in a nitrogen atmosphere obtained a similar effect to the surface treatment. At 300 °C, the CF surface was exposed as the sizing agent and impurities were removed by thermal energy. As the area in contact with the atmosphere increased, a bond of O=C-O was formed and increased, and the C=O and C-O present on the surface were combined with nitrogen atoms and removed as NO and NO_2_, as shown in Equation (2), and C=O and C-O decreased. At 500 °C, as shown in Equation (3), a small amount of oxygen atoms in the nitrogen gas were combined with the C=O and C-O on the CF surface to be removed and reduced to O_2_, and the O=C-O bond was slightly increased. At 1000 °C, C=O and C-O were combined with nitrogen atoms on the CF surface to be removed as NO and NO_2_, and oxygen atoms were removed as CO, CO_2_, and O_2_ from the bonds between carbons. From this, C=O, C-O, and O=C-O were decreased, and the CF surface was damaged due to defects inside the CF, resulting in a decrease in mechanical properties. In contrast, during heat treatment in an oxygen atmosphere, at 300 °C, oxygen atoms reacted with the carbon inside the CF and were removed as CO and CO_2_, which penetrated the surface of the CF to increase C=O, C-O, and O=C-O. It was likewise reported that O=C-O increased due to the continuous penetration of oxygen atoms, but mechanical properties were lost due to the defects in the CF from 500 °C [30]. In this study, oxygen atoms reacted with carbon at a temperature of 300 °C during thermal desizing in a nitrogen atmosphere, but the shape was maintained without defects on the fiber surface, even at 500 °C, where damage to the CF surface appeared in an oxygen atmosphere. This outcome is thought to be due to the fact that the bond between carbons was not broken down because there was almost no amount of oxygen penetrating the CF. At 1000 °C, excessive heat energy was transferred to the CF, and it was established that the reaction between the carbon existing on the surface and the oxygen in the nitrogen atmosphere affected the surface damage and loss of mechanical strength.
(2)$$C–O+N\;\to \;C+\mathrm{NO},{\;\mathrm{NO}}_{2}↑$$
(3)$$C–O+O\;\to \;C+O_{2}↑$$

## 4. Conclusions

In this study, chemical and thermal treatments were performed on rCF, and the surface of the CF was analyzed according to the desizing process parameters to confirm the mechanical and chemical properties. Moreover, the chemical state change and mechanism were identified. A change in the CF surface was observed depending on the desizing time and temperature with acetone, which is a chemical treatment method, and no significant difference was found compared to commercial CF. The tensile strength, modulus, and elongation also showed similar values. As a result of the TGA, treatment with acetone at 60 °C for 0.5 h was identified to be the optimal condition for the complete removal of the sizing agent. A look at the functional group change indicated that, compared to commercial CF, the carbonyl group (C=O) and hydroxyl group (C-O) decreased due to the removal of the sizing agent on the surface via the acetone treatment at a temperature of 25–60 °C. In addition, at 25 °C, the lactone group (O=C-O) increased due to the combination of the remaining sizing agent and oxygen, and then decreased at 60 °C as the sizing agent was completely removed. The surface energy also showed a slight decrease. This is considered to have been due to the removal of the sizing agent on the surface of the CF when treated with acetone and the decrease in the amount of oxygen, causing a decrease in the surface free energy and an increase in the contact angle.

On the other hand, when thermal desizing in an inert gas nitrogen atmosphere, there was no surface change in the CF up to 500 °C, and the tensile properties were similar to those of commercial CF. At 1000 °C, the tensile strength and elongation decreased by approximately 70% due to surface degradation. In the functional group change, at 300 °C, the C=O and C-O present on the CF surface combined with nitrogen atoms and were removed as NO and NO_2_, resulting in a decrease in C=O and C-O. At 500 °C, the oxygen atoms and C=O and C-O on the CF surface were combined and removed as O_2_, and C=O and C-O were reduced. At 1000 °C, the nitrogen and oxygen atoms and C=O and C-O on the surface of the CF reacted actively and were removed as NO, NO_2_, and O_2_. As the amount of oxygen on the surface of the CF decreased, the surface energy also decreased.

Through this study, we established the optimal conditions for the desizing process and identified differences in the mechanisms during chemical and thermal desizing treatment. In the future, we will conduct optimal surface treatment and resizing after desizing treatment to optimize the upcycling process of rCF. We plan to establish a process and manufacture and evaluate carbon composites using thermoplastic and thermosetting resins and upcycled rCF, thereby contributing to the commercialization of automobile parts using rCF.

## Figures and Tables

**Figure 1 materials-16-06732-f001:**
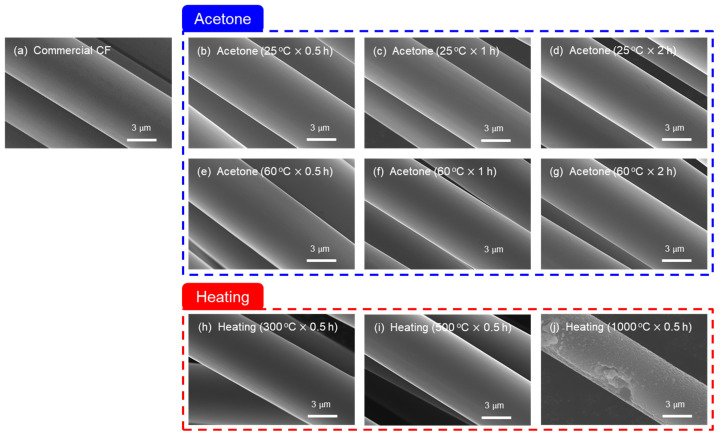
SEM image of carbon fibers with desizing process conditions: (**a**) commercial CF, (**b**) 0.5 h, (**c**) 1 h, (**d**) 2 h acetone at 25 °C (**e**) 0.5 h, (**f**) 1 h, and (**g**) 2 h acetone at 60 °C, (**h**) 300 °C, (**i**) 500 °C, (**j**) 1000 °C heating 0.5 h with nitrogen atmosphere.

**Figure 2 materials-16-06732-f002:**
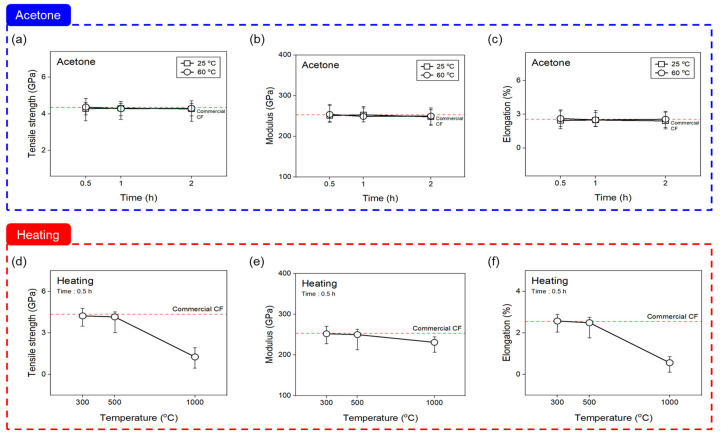
Changes in tensile properties of carbon fiber according to desizing process conditions: (**a**–**c**) acetone, and (**d**–**f**) heating with nitrogen atmosphere. The red line shows the average value for commercial carbon fiber.

**Figure 3 materials-16-06732-f003:**
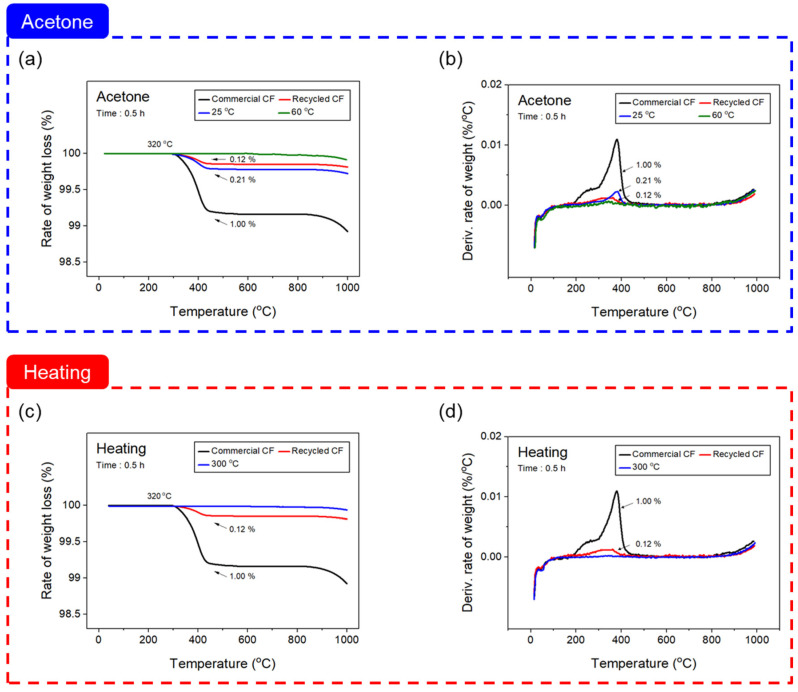
Thermogravimetric analysis of carbon fiber according to the desizing process conditions: (**a**,**b**) acetone, and (**c**,**d**) heating with nitrogen atmosphere.

**Figure 4 materials-16-06732-f004:**
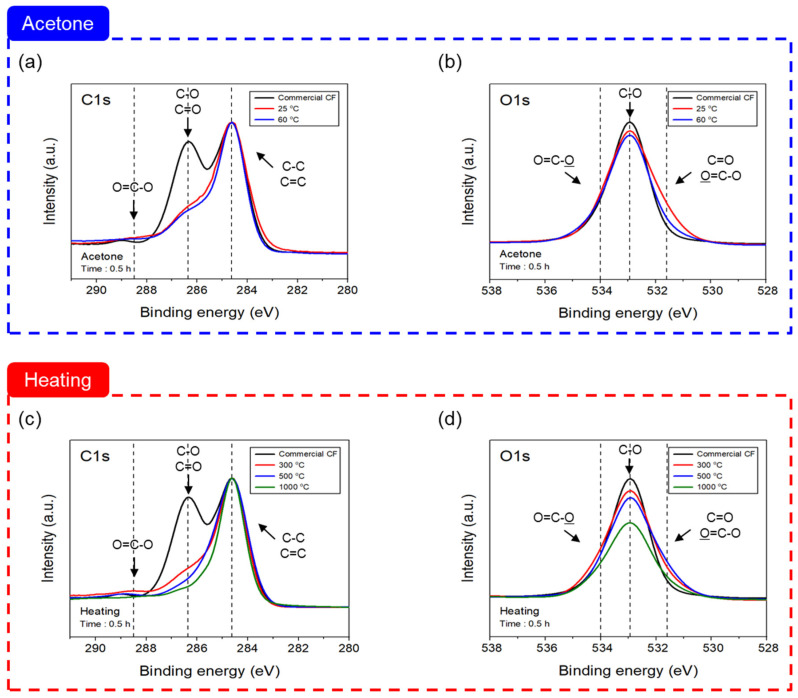
C1s and O1s XPS spectra of carbon fiber depending on desizing process conditions: (**a**,**b**) acetone, and (**c**,**d**) heating with nitrogen atmosphere.

**Figure 5 materials-16-06732-f005:**
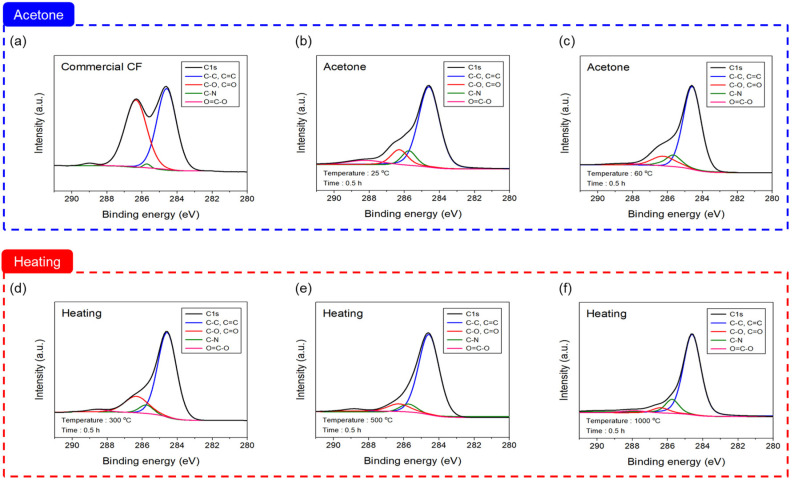
XPS spectra of C1s after desizing process conditions: (**a**) commercial CF, (**b**,**c**) acetone, and (**d**–**f**) heating with nitrogen atmosphere.

**Figure 6 materials-16-06732-f006:**
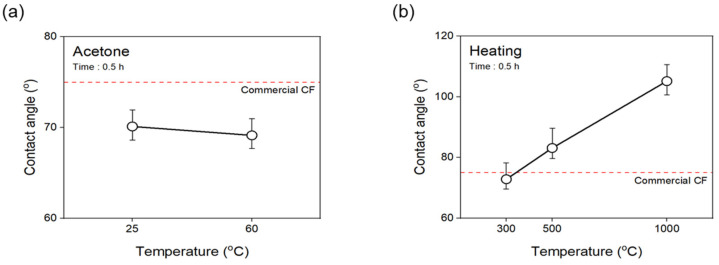
Contact angle change according to desizing process conditions: (**a**) acetone, and (**b**) heating with nitrogen atmosphere. The red line shows the average value for commercial carbon fiber.

**Figure 7 materials-16-06732-f007:**
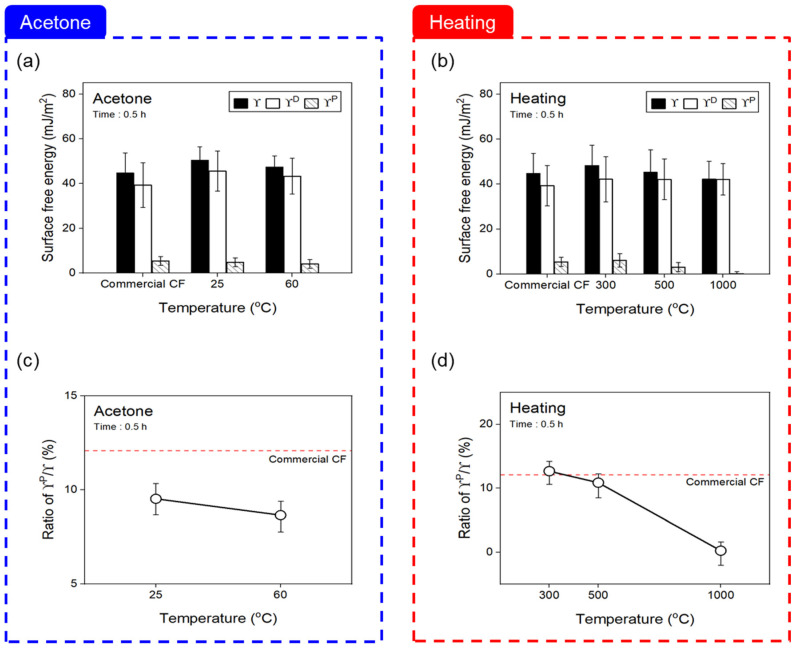
Surface energy of desized carbon fibers covered with: (**a**,**c**) acetone, and (**b**,**d**) heating with nitrogen atmosphere. The red line shows the average value for commercial carbon fiber.

**Figure 8 materials-16-06732-f008:**
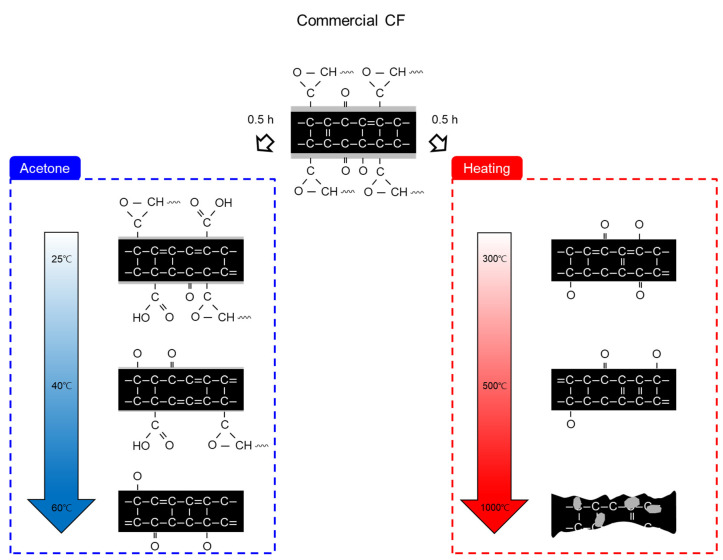
Schematic of carbon fiber depending on desizing process conditions.

**Table 1 materials-16-06732-t001:** Characteristics of carbon fiber in this paper.

Type	Commercial CF	Recycled CF
Tensile strength (GPa)	4.49	3.45
Modulus (GPa)	261	256
Elongation (%)	2.62	2.08
Density (g/cm^3^)	1.80	1.80

**Table 2 materials-16-06732-t002:** Surface element composition of carbon fiber according to desizing process conditions.

Desizing Process	Temperature	Elemental Composition (at. %)				O/C
		Carbon	Oxygen	Nitrogen	Silicon	
Commercial CF		76.31	21.31	0.75	1.63	0.28
Acetone	25 °C	77.85	18.06	1.47	2.62	0.23
	60 °C	79.83	16.46	1.88	1.83	0.20
Heating	300 °C	80.14	16.97	1.17	1.72	0.21
	500 °C	84.45	15.13	1.19	1.23	0.18
	1000 °C	86.64	9.47	0.75	3.15	0.11

**Table 3 materials-16-06732-t003:** Chemical groups of desizing process conditions.

Desizing Process	Temperature	C1s (at. %)			
		C-C, C=C	C-O, C=O	C-N	O=C-O
Commercial CF		50.85	47.10	1.98	0.07
Acetone	25 °C	68.35	18.10	6.88	6.67
	60 °C	73.57	16.36	7.95	2.12
Heating	300 °C	77.37	19.25	1.46	1.92
	500 °C	85.65	10.05	1.39	2.91
	1000 °C	84.15	5.43	9.01	1.41

## Data Availability

The data presented in this study are available on request from the corresponding author.

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
