# Peer review of "The Effect of Chemical and Thermal Treatment for Desizing on the Properties and Chemical Functional Groups of Carbon Fiber"

_materials, 2023, doi:10.3390/ma16206732_

Round 1

Reviewer 1 Report

In this study, the surface of carbon fibers and recycled carbon fibers was analyzed by chemical and heat treatment, and their mechanical and chemical properties were confirmed according to the desizing process parameters. Under the optimal conditions of desizing treatment rCF, carbon composites were manufactured with the best surface treatment and sizing treatment conditions, and the upgrading cycle of carbon composites was optimized.

The structure of the article is reasonable, the logic is clear, and the experimental design is in line with reality. However, this article needs to make the following modifications before publication:

The reviewer suggests that the authors highlight the value and significance of this study in the final part of the abstract section.

The basic principles of some experimental designs in the study are not explained in detail. The principle needs to be further explained so that other readers can have a clearer understanding of the experimental process.

The point of view in line 145-146 is 'During heat treatment. The atmosphere can be seen to have a greater effect on the surface reaction of CF than the heat treatment temperature. It is determined that the energy that damages the bond is small. ' Can a set of experimental data be added to enhance credibility.

The changes in the data can be seen in the figure, but it is recommended to add key information or tags, so that readers can more intuitively and quickly identify the information expressed by the author.

The conclusion part should be more refined to make the findings and contributions of the paper clearer. Furthermore, please note the difference between the conclusions and abstract.

Reviewer 2 Report

The study is interesting and the reviewer believes that the manuscript can be published in this journal. 

However there are some points that the reviewer suggest to improve:

- Regarding the mechanical tests for the fiber degraded at 1 000 ºC the section was reduced, the strength determined need to be updated for this new section. 

- How do you explain to the fact of the contact angle increase when degraded the fibers at high temperature. This is not the clear in the manuscript. 
